# The Rise of China and Evolving Defense Cooperation between India and Japan

**Keerthiraj** [1,*] and **Takashi Sekiyama** [2]

1  Alliance School of Liberal Arts, Alliance University, Bengaluru 562106, India
2  Graduate School of Advanced Integrated Studies in Human Survivability, Kyoto University, Kyoto 606-8306, Japan; sekiyama.takashi.2e@kyoto-u.ac.jp
*  Correspondence: krj492@gmail.com

**Abstract:** This research paper investigates the evolving defense cooperation between India and Japan in the context of the strategic landscape in the Indo-Pacific region, particularly considering China's rise. The existing literature recognizes China's increasing influence as a significant factor in India-Japanese security cooperation, but tends to overlook the dynamics behind India's shifting stance. The study reveals that India initially hesitated to actively engage in anti-China security cooperation with Japan until the mid-2010s, but later adopted a more proactive partnership. An analysis of policy documents, political speeches, and government records attributes India's initial reluctance to its commitment to non-alignment and strategic autonomy, its desire to balance relations with both China and Japan, and its domestic political priorities favoring economic development over military expansion or strategic alliances. However, growing concerns over China's assertiveness, military modernization, a change in political leadership, and the Quad's development as a regional security platform have prompted India's attitude shift. The research's recommendations not only offer a roadmap for India, Japan, and other Indo-Pacific countries with deep economic ties to China, but also help combat China's military threats to contribute to regional stability and security, address common challenges, and foster a peaceful and prosperous Indo-Pacific region.

**Keywords:** India-Japan defense cooperation; Indo-Pacific; regional stability; strategic autonomy; balance of power; the rise of China

## 1. Introduction

As the global landscape undergoes dramatic changes, the evolving defense cooperation between nations becomes a critical aspect of international relations. In particular, the rise of China as a dominant power has raised concerns and challenges for its neighbors, prompting them to seek strong partnerships to ensure regional stability and security. On the other hand, the neighboring countries have deep economic ties with China. It is a difficult challenge for the Indo-Pacific countries to curb Beijing's expansionism while maintaining good economic relations with China. Japan and India are also countries that are bound by deep economic ties with China, but also face military threats. China is the largest trading partner for both Japan and India. On the other hand, China's assertive behavior and rapid military modernization have alarmed countries in the Indo-Pacific region, encouraging them to explore new strategic partnerships to balance Beijing's power (Fravel 2018). The India-Japanese security cooperation has evolved in response to this changing regional environment, with both nations recognizing the need for a partnership based on shared values, interests, and mutual respect. Particularly, India and Japan have been growing their defense cooperation since the mid-2010s.

The evolution of the defense cooperation between India and Japan has been examined by various scholars who have assessed the motivations, challenges, and opportunities inherent in this comprehensive partnership. The literature highlights the growing importance of

the India-Japanese defense cooperation as a response to the changing regional security environment, particularly in the context of China's rise. The partnership between the two countries is seen as a crucial component of their broader strategic objectives, which are aimed at promoting regional stability and ensuring a rules-based order in the Indo-Pacific region (Auslin 2016; Liff 2015; Pant 2007; Naidu 2007; Brewster 2010a; Naidu and Yasuyuki 2019; Fravel 2018; Brewster 2013; Brewster 2014; Kinne and Bunte 2020; Chakraborty 2020; Burgess and Beilstein 2018; Harner 2014; Panda 2012). While the existing literature acknowledges China's rising influence as a major background for the India-Japanese security cooperation, it appears to overlook the dynamics behind India's changing stance towards this partnership. Specifically, there is a research gap concerning the reasons for India's initial reluctance to engage actively in anti-China cooperation with Japan until the mid-2010s, despite signing various agreements. Furthermore, the factors that led to India's shift in position and increased proactiveness in its security partnership with Japan remain inadequately explored. Addressing these questions will provide valuable insights into the nuances of the India-Japanese security cooperation and contribute to a more comprehensive understanding of the evolving relationship between these two nations.

Why did Japan and India promote their security cooperation against China despite their deepening economic dependence on the country? Did Japan and India, both of which have deep economic ties with China, experience any reluctance regarding China when promoting their mutual security cooperation? Why did India shift its position and become more proactive in its security partnership with Japan after the mid-2010s? How have these changes in India's stance towards its partnership with Japan influenced the broader India-Japanese security cooperation and regional dynamics? This paper aimed to answer these questions with a particular focus on India's policy stance. This paper examined the various dimensions of the evolving India-Japanese defense cooperation, highlighting its implications for regional security and global geopolitics. It explored the motivations, challenges, and opportunities inherent in this comprehensive partnership, and assessed how it contributed to shaping the Indo-Pacific's strategic balance. By situating the India-Japanese security cooperation within the broader context of China's rise and the shifting dynamics of the Indo-Pacific region, this paper underscores its significance as a key pillar of regional stability and prosperity.

## 2. Materials and Methods

The research methodology relied predominantly on primary sources, such as policy documents, political speeches, and government records, to investigate the facts about India's shifting position regarding its security cooperation with Japan. These primary sources provided the foundation for the study's arguments, ensuring that the analysis is grounded in concrete evidence. The facts found through a review of primary sources are presented in the results section. Secondary sources, including academic papers and media articles, were used to supplement the primary sources and provide additional context for the analysis. The analysis based on the review of secondary sources is presented in the discussion section. The research employed a qualitative approach, analyzing the gathered evidence to address the research questions and objectives without necessarily formulating hypotheses. This approach enabled a thorough examination of the factors influencing India's changing stance and its impact on the broader India-Japanese security cooperation and regional dynamics.

## 3. Results

### 3.1. Tokyo's Positive Attitude towards Japan-India Cooperation

Former Japanese Prime Minister Shinzo Abe's strategic vision of the "confluence of two seas" during his visit to India in 2006–07, emphasized the natural partnership between Japan and India (MOFA Japan 2007). The shared vision has since evolved into the alignment of Japan's Open and Free Indo-Pacific Strategy and India's Act East Policy, reflecting Tokyo's increasing commitment to India amidst geopolitical shifts (MOFA



Japan 2017). Japan's positive attitude towards India is substantiated by significant economic agreements, such as the Mumbai-Ahmedabad High-Speed Rail (MAHSR) project. This project, financed mainly by Japan through long-term, low-interest loans, underscores Japan's investment in India's development. In the realm of defense and security, Tokyo's commitment to its cooperation with Delhi is evidenced by agreements including the potential purchase of Japan's US-2 amphibious plane by India. Moreover, the two nations' participation in the annual MALABAR naval exercises alongside the U.S. cements this burgeoning defense cooperation. The progress in civil nuclear energy cooperation and the initiation of the "two-plus-two" Foreign and Defense Ministerial Meeting in 2019 further exemplify Japan's proactive stance in bolstering the Japan-India alliance. The meeting was a pivotal advancement, upgrading dialogues previously held at the secretary level and showcasing Tokyo's preference for this collaborative mechanism (Thakker 2019). This trajectory was also marked by the negotiation of the Acquisition and Cross-Servicing Agreement (ACSA), further reinforcing the military engagement between the two countries. The prospective agreement highlights Tokyo's commitment to deepening its security ties with India, which underpins Japan's positive foreign policy towards India and their shared regional stability interests.

The cooperation between Japan and India began in the 2000s. When India conducted a nuclear test in 1998, Japan imposed severe sanctions on India. Japan, however, began to promote practical security cooperation while continuing to lobby India on the nuclear issue. In August 2000, while Japan's sanctions against India's nuclear tests were still in place, Prime Minister Yoshiro Mori visited India and began rebuilding Japan-India relations. This was indeed the first visit by a Japanese prime minister in ten years, since Prime Minister Toshiki Kaifu's visit to India in 1990. Prime Minister Mori's visit marked a turning point in improving Japan-India relations, which had been estranged by India's nuclear tests, and advancing defense cooperation. Prime Minister Mori and Prime Minister Atal Bihari Vajpayee agreed to establish the "India-Japan Global Partnership for the 21st Century". The two leaders agreed that it was important to hold annual security dialogues and defense authorities' meetings to address the full range of issues of mutual concern, including nuclear disarmament and non-proliferation (Ministry of Foreign Affairs, Japan 2001).

By the mid-2000s, Japanese diplomats increasingly regarded India as a strategic partner (Hirose 2007). As one concrete example, Japan's Ministry of Foreign Affairs (MOFA) made institutional changes with an emphasis on India, establishing a new "Southern Asia Division" in the Asia-Pacific Bureau to oversee South Asia and Southeast Asia. In April 2005, Prime Minister Junichiro Koizumi visited India to advance Japan-India strategic cooperation. Against the backdrop of the rapid deterioration of Japan-China relations at that time, this visit by Koizumi to India opened an era of comprehensive security cooperation between Japan and India, with an emphasis on regional security. In April 2005, Prime Minister Junichiro Koizumi visited India to advance Japan-India strategic cooperation. Against the backdrop of the rapid deterioration of Japan-China relations at the time, this visit by Koizumi to India opened an era of comprehensive security cooperation between Japan and India, with an emphasis on regional security (Ministry of Foreign Affairs, Japan 2005). During the Abe administration, Tokyo and New Delhi developed further defense and security cooperation. Prime Minister Abe was convinced of the importance of India as a partner of Japan from a global strategic perspective after seeing Prime Minister Koizumi's diplomatic relations with China and South Korea deteriorate over historical issues (Abe 2021). In December 2006, Prime Minister Manmohan Singh visited Japan and held a Japan-India summit meeting with Prime Minister Shinzo Abe. At this summit meeting, a Japan-India Joint Declaration was issued, upgrading the Japan-India relationship to a "strategic global partnership". Of particular significance was the Joint Declaration's decision to institutionalize the Strategic Dialogue between the Foreign Ministers and regular policy dialogue between the Advisor to the Prime Minister on the National Security of In-

dia and his Japanese counterpart (5). In other words, Japan and India agreed to initiate an annual defense ministers' meeting, in addition to annual summit-level reciprocal visits.

In August 2007, Prime Minister Abe visited India to help strengthen security and defense cooperation. Prime Minister Abe delivered a speech titled "The Intersection of Two Seas" in the Indian Parliament, in which he argued that Japan and India should cooperate in securing sea lanes as maritime nations. Abe also positioned the Japan-India relationship as "the most potential bilateral relationship in the world". In the Japan-India Joint Statement at the time, the two countries laid out a "roadmap" for a strategic global partnership and confirmed the holding of a ministerial-level inter-foreign strategic dialogue and an annual defense/defense ministerial meeting (Ministry of Foreign Affairs, Japan 2007). In October 2008, Prime Minister Singh visited Japan and met with Prime Minister Aso to announce the "Joint Declaration on Japan-India Security Cooperation". This Joint Declaration was of great significance for Japan-India relations because India was the third country, only after the United States and Australia, to which Japan had issued a Joint Declaration on Security Cooperation. The Joint Declaration institutionalized a multilayered framework for a number of dialogues, consultations, and exchanges on diplomatic, defense, and security cooperation (Ministry of Foreign Affairs, Japan 2008).

In October 2010, Prime Minister Manmohan Singh visited Japan and held a summit meeting with Prime Minister Naoto Kan. The leaders issued a joint statement called "A Vision for a Japan-India Strategic Global Partnership for the Next Decade". They welcomed the steady implementation of the "Action Plan" for Japan-India security cooperation and agreed to strengthen it. In July of the same year, they launched the Foreign and Defense "2+2" Dialogue at the vice-ministerial level. When the second Abe administration took office in December 2012, he accelerated the strengthening of Japan-India relations on his own initiative. In May 2013, Prime Minister Manmohan Singh visited Japan and held a summit meeting with Prime Minister Abe. In the joint statement, "Strengthening the India-Japan Strategic Global Partnership Beyond the 60th Anniversary of the Establishment of Diplomatic Diplomatic Relations," the two countries expressed "expectations for further promotion of dialogue and cooperation on maritime issues" and agreed to "further enhance and promote anti-piracy operations and bilateral joint training between navies". In December 2013, Japan formulated its first "National Security Strategy," which identified India as an "important regional partner" that shares "universal values, strategic interests and concerns." The strategy also identified India as "a geopolitically important country located in the middle of our sea lanes with growing influence due to its population, which is expected to become the largest in the world, high economic growth and potential economic power," and called for strengthening relations in a wide range of areas, including maritime security, based on the strategic global partnership established between the two countries (Abe 2011; Cabinet Office, Japan 2013).

### 3.2. India's Initial Reluctance

In contrast to Japan's positive attitude, India was initially cautious about developing India-Japan security cooperation. The 2008 Joint Declaration on Security Cooperation between India and Japan (Ministry of Foreign Affairs, Japan 2008) underscores the mutual commitment to peace and stability in the region. However, the document does not explicitly mention China or any intention to counterbalance its influence. This highlights India's cautious approach to its security relationship with Japan and its avoidance of any overt anti-China stance during that period. In a 2009 speech by then-Indian Prime Minister Manmohan Singh, he emphasized the importance of maintaining a "strategic and cooperative partnership" with China while also pursuing stronger ties with Japan (Ministry of External Affairs, Government of India 2012). This demonstrates India's intention to strike a balance in its relationships with both countries, reflecting its hesitance to adopt an openly confrontational approach towards China.

An analysis of India's defense budget allocations during the period leading up to the mid-2010s further supports the argument of its cautious approach towards its secu-

rity cooperation with Japan. India's defense budget remained relatively modest, with an emphasis on modernizing its armed forces rather than expanding its military capabilities or entering into significant strategic partnerships (Ministry of Defence 2014). The 2010 India-Japan Comprehensive Economic Partnership Agreement (CEPA) (Ministry of Commerce and Industry 2011) was a significant milestone in the two countries' relationship, and aimed at deepening economic and trade ties. However, the agreement primarily focused on economic aspects, with no explicit mention of defense or security cooperation. This reflects India's prioritization of economic engagement over security cooperation during that period. A 2011 speech by then-Indian Foreign Secretary Nirupama Rao at the Institute for Defence Studies and Analyses (IDSA) highlighted the importance of engaging with China constructively and managing differences through dialogue (Economic Times 2011). The speech illustrates India's preference for diplomacy and negotiation over confrontation, further explaining its reluctance to actively pursue anti-China security cooperation with Japan at the time. In the India-Japan Vision 2025 document (Ministry of External Affairs, India 2015), which outlines the strategic and global partnership between the two countries, there is a notable emphasis on regional connectivity, economic cooperation, and people-to-people exchanges. Although the document refers to enhanced security cooperation, it does not explicitly mention China or the need to counterbalance its influence. This illustrates India's measured approach in advancing its security relationship with Japan while avoiding confrontation with China.

The trend of India's initial reluctance and positive trade approach towards Japan can also be observed with a larger picture of India's increasing imports from Japan from mid-2010 onwards (Figure 1).

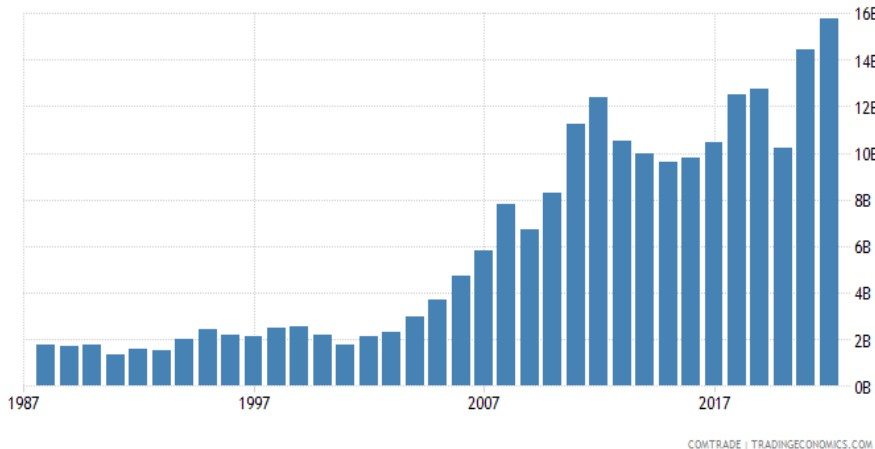

**Figure 1.** India Imports from Japan (Last 50 years). Source: Trading Economics (Trading Economics 2023).

As stated above, the review of policy documents, political speeches, and government records reveals that India has been initially reluctant to actively engage in anti-China security cooperation with Japan until the mid-2010s. The factors behind its reluctance discussed later can be attributed to its adherence to the principles of non-alignment and strategic autonomy, its desire to maintain a balance in its relations with both China and Japan, and its domestic political considerations that prioritized economic development over military expansion or strategic alliances.

*3.3. India's Shift in Position and an Assertive Foreign Policy*

In 2014, Indian Prime Minister Narendra Modi and Japanese Prime Minister Shinzo Abe agreed to upgrade their bilateral relationship to a "Special Strategic and Global Partnership" during Modi's visit to Japan (Ministry of External Affairs, Japan 2014). This agreement marked a significant step in deepening the security cooperation between the two countries and indicated India's willingness to engage more proactively with Japan in

response to regional security challenges. The 2016 India-Japan Civil Nuclear Cooperation Agreement (Ministry of External Affairs 2016) represents another example of India's growing security cooperation with Japan. This agreement facilitated the exchange of nuclear technology, equipment, and expertise for peaceful purposes, signaling India's readiness to collaborate more closely with Japan on strategic issues. During the 2018 Shangri-La Dialogue, Indian Prime Minister Narendra Modi emphasized the importance of a "free, open, and inclusive" Indo-Pacific region (Modi 2018). This speech highlighted India's commitment to regional security and further underscored the growing importance of its security partnership with Japan in achieving this vision. Additionally, the development of the Quadrilateral Security Dialogue (Quad) between the United States, Australia, India, and Japan in 2017 provided a multilateral platform for India to deepen its security cooperation with Japan and other like-minded countries (Ministry of External Affairs 2017). The Quad's focus on promoting a free, open, and inclusive Indo-Pacific region aligned with India's strategic interests, further motivating India to enhance its security partnership with Japan (Grare 2018). The 2020 India-Japan Acquisition and Cross-Servicing Agreement (ACSA) negotiations marked another milestone in the security partnership between the two countries (Laskar 2020; Siddiqui 2020). The ACSA, signed in 2020, enabled the reciprocal sharing of logistics, supplies, and services between the armed forces of India and Japan, thus expanding their ability to cooperate on regional security issues (Ministry of Foreign Affairs, Japan 2020). As shown above, these additional policy documents, political speeches, and government records substantiate India's shift towards a more proactive security partnership with Japan since the mid-2010s, which was driven by changes in political leadership, regional security concerns, and the development of multilateral platforms, such as the Quad.

### 3.4. *China's Rising Influence and Evolving Defense Cooperation between India and Japan*

The third scope of this paper is to comprehend the implications of the evolving defense cooperation between India and Japan, particularly in the context of China's rising influence. In this regard, the India-Japan Joint Declaration on Security Cooperation (Ministry of Foreign Affairs, Japan 2008) highlights both countries' commitment to peace, stability, and prosperity in the Indo-Pacific region. This policy document serves as a foundation for their ongoing collaboration in areas such as counterterrorism, maritime security, and regional capacity-building. In 2015, the India-Japan Vision 2025 document (Ministry of External Affairs, India 2015) emphasized the need for deeper defense cooperation, including joint exercises, defense equipment, and technology cooperation, and the sharing of best practices. This policy document demonstrates the shared strategic interests of both countries and their willingness to work together to address regional security challenges.

During his visit to Japan in 2016, Indian Prime Minister Narendra Modi stated that a "strong India-Japan partnership is not only in the interest of the two countries but is also important for peace and security in the region" (Prime Minister's Office, India 2016). This statement underscores the broader implications of the India-Japan defense partnership for regional stability. The 2019 India-Japan 2+2 Ministerial Dialogue (PIB, India 2019) saw the two countries' foreign and defense ministers discussing ways to further strengthen their security cooperation, including through capacity-building initiatives, joint exercises, and defense technology collaboration. The establishment of this high-level dialogue mechanism signifies the deepening of the India-Japan defense partnership and its potential impact on regional security dynamics. These additional policy documents, political speeches, and government records further support the argument that the evolving defense cooperation between India and Japan has significant implications for regional stability, the balance of power in the Indo-Pacific region, and the development of joint capabilities to address common security challenges.

## 4. Discussion

### 4.1. Possible Factors behind Japan's Positive Attitude

In the face of a shifting regional balance of power, particularly with China's rise and the perceived relative decline of the United States, Japan's broader security strategy has driven its proactive approach towards India (Johnston 2013; Rachman 2011). This strategy is centered around enhancing its national defense capacity, solidifying the US-Japan alliance, and leveraging relationships with like-minded nations, such as India, to potentially develop diplomatic and military alliances. With concerns mounting over China's growing diplomatic sway over ASEAN, Japan was insistent on incorporating Australia, India, and New Zealand into the East Asia Summit (EAS). This maneuver was aimed at thwarting China's bid to confine the EAS membership to the ASEAN Plus Three framework (China, Japan, and South Korea), which would enable China to retain its regional influence.

Japan's policy towards India was also influenced by fluctuations in US engagement in Asia. Domestic debates in the US raised the specter of a potential relative retrenchment from international affairs (Posen 2013; Nordin and Weissmann 2018). This instigated apprehensions that China might exploit periods of low US engagement in Asia to its advantage (Brooks et al. 2013). In 2016, two significant events highlighted China's growing influence in East Asia and beyond: the emergence of the Belt and Road Initiative (BRI), and China outbidding Japan on a project in Indonesia through a competitive financial scheme (Permanent Court of Arbitration 2016). Such developments underscored China's ascending clout and presented strategic challenges for Japan's positioning in the region (Asian Development Bank 2007). Yet, Japan's positive policy towards India is not merely reactive to the structural changes in Asia's balance of power. Internal factors have also contributed, including the enduring US-led international order and the expansive strategic thinking of Prime Minister Abe. Abe envisaged an international order rooted in 'universal values,' which could bind democratic states and impose normative constraints on non-democratic states, including China.

Why has Japan-India security cooperation been steadily deepening? It may be because there are factors that encourage the deepening of cooperation, while there are no factors that hinder it. What, then, are the factors driving the deepening? It is worth noting that the evolution of the security relationship between Japan and India began in the late 2000s, when Japan-China relations deteriorated. In 2000, Japan-China relations deteriorated due to anti-Japanese riots and the China Risk. The results of public opinion polls show a change in the Japanese public's attitudes towards China (Figure 2). In the 1980s, approximately 70% of the Japanese public felt a sense of affinity towards China. However, this dropped to approximately 50% in the wake of the Tiananmen Square incident in 1989. In the 2000s, anti-China sentiment in Japan increased further, triggered by anti-Japanese demonstrations in China (2003) and the 2010 collision between Japanese and Chinese vessels off the Senkaku Islands. Reflecting a strong anti-Chinese sentiment, Japan terminated its official development assistance (ODA) loans to China in 2008 (Sekiyama 2012). It was also the time when the U.S. policy changed accordingly. In November 2011, U.S. President Barack Obama declared a shift in the U.S. global strategy towards "a larger and long-term role in shaping the Asia Pacific by upholding close partnership with its allies and friends" in a speech to the Australian Parliament during a visit (White House 2011). This was a historic speech that made it clear both domestically and internationally that the U.S. had shifted from its traditional policy of engagement to a policy of deterrence against an expanding China. This meant that for the first time in about 40 years, since Richard Nixon's visit to China, U.S.-China relations had once again entered an era of confrontation. President Obama also said in his speech that the United States welcomed India to play a larger role as an Asian power. Under this situation, closer U.S.-India relations also progressed. The fact that the United States, Japan's most important ally, has taken a more confrontational stance towards China and begun to move closer to India has lowered the hurdle for Tokyo to pursue Japan-India security cooperation (Bajpai 2019).

As the exisiting literature highlights, the growing importance of India-Japan defense cooperation can be understood as a response to the changing regional security environment, particularly in the context of China's rise. The partnership between the two countries is seen as a crucial component of their broader strategic objectives, which are aimed at promoting regional stability and ensuring a rules-based order in the Indo-Pacific region (Auslin 2016; Liff 2015; Pant 2007; Naidu 2007; Brewster 2010b, 2013, 2014; Naidu and Yasuyuki 2019; Fravel 2018; Kinne and Bunte 2020; Chakraborty 2020; Burgess and Beilstein 2018; Harner 2014; Panda 2012).

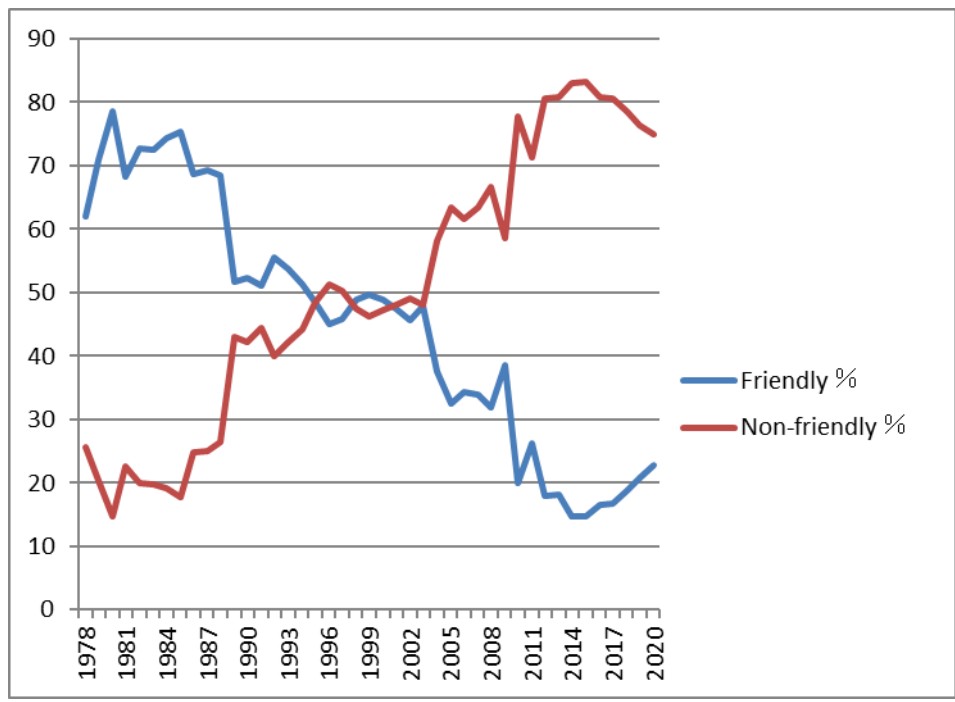

**Figure 2.** Japan's Sense of Affinity towards China. Source: Cabinet Office Poll on Diplomacy (Cabinet Office 2022).

### 4.2. Possible Factors behind India's Initial Reluctance

Why, despite Japan's positive attitude, was India initially cautious about developing India-Japan security cooperation until the early 2010s? One factor that may have contributed to India's reluctance is its traditional adherence to the policy of non-alignment and strategic autonomy (Muraviev et al. 2022). By avoiding entanglement in formal alliances, India sought to maintain its independence and flexibility in foreign policy matters (Ganguly and Pardesi 2009). Consequently, India's preference for strategic autonomy might have limited its willingness to actively engage in anti-China security cooperation with Japan, as their collaboration could be perceived as a departure from this policy. Another factor that could have influenced India's cautious approach is its complex relationship with China, which encompasses elements of both competition and cooperation (Paul 2018). India has long engaged in economic and diplomatic interactions with China, despite the strategic rivalry and unresolved border disputes.

Even if the possibility of an official alliance between India and Japan appears advantageous in terms of bolstered security cooperation and increased regional stability, potential adverse conditions of this alliance must be taken into consideration. First of all, an official alliance might jeopardize India's long-standing policy of strategic independence. Furthermore, such an alliance may seriously deteriorate India's relations with China. A formal alliance could be viewed as a provocative action by China, given the territorial disputes and previous hostilities between China and both India and Japan. On the other hand, it might also make things more difficult for India's strategic partner and major defense sup-

plier, Russia. Because of Russia's close links to China, a formal alliance between India and Japan might be perceived as a negative step. In a larger geopolitical sense, a regional domino effect might be set off by an Indo-Japan alliance, resulting in worsened security issues and an arms race in the Indo-Pacific area. Other regional players may retaliate by forming their own coalitions or increasing their military power, which would make the regional security climate more turbulent and unpredictable. India's initial reluctance to actively cooperate with Japan in security matters might have stemmed from its desire to avoid provoking China and to preserve a semblance of balance in its relations with both nations (Mathur 2012). Moreover, domestic political considerations may have played a role in India's initial hesitance. Until 2014, the Indian National Congress-led government pursued a relatively cautious approach to India's foreign policy, prioritizing a soft power image (Pant 2012).

### 4.3. Possible Factors behind India's Shift in Position

Why, then, did India change its stance in the mid-2010s despite the factors examined above? One significant factor contributing to India's shift in position is the growing assertiveness of China in the Indo-Pacific region, particularly in the South China Sea and along the Sino-Indian border (Brewster 2017). China's increasingly aggressive behavior, coupled with its rapid military modernization, may have prompted India to reassess its security strategy and seek closer ties with Japan to counterbalance China's influence in the region (Basrur and Narayanan Kutty 2022).

Another factor that could have influenced India's shift is the change in its political leadership with the election of Prime Minister Narendra Modi in 2014. The election of Prime Minister Narendra Modi and his Bharatiya Janata Party in 2014 signaled a shift in India's foreign policy, with a greater emphasis on national security and strategic partnerships (Basrur 2017). This transition in political leadership and priorities might have contributed to India's increased willingness to engage with Japan on security matters. Modi's administration placed a greater emphasis on national security and strategic partnerships, leading to a more assertive foreign policy. Under Modi's leadership, India pursued a more proactive approach towards security cooperation with Japan, as evidenced by the signing of the India-Japan Vision 2025 document (Ministry of External Affairs, India 2015).

Issues beyond the Sino-Indian border, such as maritime security and counterterrorism, have also been instrumental in driving India towards closer ties with Japan (Bradford 2021). With the Indian Ocean region becoming a hotbed of piracy and maritime disputes, Japan and India have recognized the need for cooperation in maintaining regional maritime security. Their shared interests in the free and open navigation in the Indo-Pacific region have motivated them to deepen their defense and security cooperation (Choong 2019; Keerthiraj and Devaiah 2022; Siahaan and Risman 2020; Toan et al. 2023). Similarly, shared concerns over terrorism, specifically the rise of extremist elements in the Indo-Pacific region, have brought India and Japan closer together. The need for a cooperative approach to counterterrorism has been acknowledged by both nations, leading to a deepening of ties in security and intelligence sharing (Koga 2019; Naidu and Yasuyuki 2019). The dynamics of India-Japan security cooperation have changed because of India's realignment of its foreign policy, external pressure from China's assertiveness, and internal causes, such as piracy and terrorism. The consequences of this change will have a significant impact on the power dynamics in the Indo-Pacific region.

### 4.4. Implications of the Growing India-Japan Defense Cooperation

One of the implications of the growing India-Japan defense cooperation is the potential for enhanced regional stability and a balance of power in the Indo-Pacific region. The strengthening of the India-Japan partnership, along with their involvement in the Quad, contributes to the development of a regional security architecture aimed at maintaining peace and promoting a rules-based order (Grare 2018). The deepening India-Japan defense partnership may also encourage other regional actors to pursue similar security co-

operation initiatives, leading to a network of partnerships that can collectively address regional security challenges (Vinod 2021). For instance, the trilateral dialogues between India, Japan, and Australia, as well as between India, Japan, and the United States, illustrate the growing convergence of strategic interests among regional powers in response to China's rise (Smith 2020).

Another implication is the potential impact on India's relations with China. While India's closer ties with Japan may be perceived as a containment strategy against China, it is important to note that India has consistently pursued a policy of strategic autonomy and maintained a multi-vector approach to its foreign relations (Pardesi 2022, p. 52). Therefore, the evolving India-Japan defense cooperation should not necessarily be seen as an anti-China alliance, but as a natural progression of India's strategic partnerships. The India-Japan defense partnership may also contribute to the development of joint capabilities and the sharing of technology, particularly in areas such as maritime security and intelligence sharing (Chaulia 2018). This cooperation can not only enhance the security capabilities of both countries, but also foster greater interoperability between their armed forces. The evolving defense cooperation between India and Japan has significant implications for regional stability, the balance of power in the Indo-Pacific region, and the development of joint capabilities to address common security challenges.

### 4.5. Policy Analysis and Suggestions

Based on the analysis stated so far, some suggestions and recommendations can be proposed for the future defense cooperation between India and Japan. Since the growing assertiveness of China in the Indo-Pacific region, particularly in the South China Sea, was a significant factor contributing to the development of India-Japan defense cooperation, the two countries should deepen their cooperation in several ways. First, India and Japan can benefit from increased cooperation in defense technology, equipment, and research and development. This could include the joint development of defense platforms, such as unmanned systems, missile defense, and cyber capabilities, as well as the sharing of best practices and know-how in defense production and technology. Second, to better respond to regional security challenges, India and Japan should work towards enhancing the interoperability of their armed forces through joint exercises and personnel exchanges, and harmonizing their military doctrines and communication protocols. Third, given the strategic significance of the Indo-Pacific region, India and Japan should further enhance their maritime security cooperation by conducting joint naval exercises, sharing intelligence on maritime threats, and collaborating on capacity-building initiatives for regional partners. On the other hand, it is clear that both India and Japan want to avoid provoking China and preserve a semblance of balance in their relations with both nations. While deepening their defense cooperation, the two countries should continue to pursue balanced relations with other regional powers, including China, to avoid exacerbating regional tensions and encourage constructive dialogue on regional security issues.

Moreover, just as India was reluctant to engage in security cooperation with Japan until the mid-2010s, the policy of non-alignment and strategic autonomy is still important for India. Thus, it is important for India to maintain a balance through multilateral cooperative relationships, rather than promoting its security affairs solely with Japan. This could be possibly true for Japan also. Therefore, India and Japan, along with the United States and Australia, should continue to strengthen the Quad framework by promoting a free, open, and inclusive Indo-Pacific region, addressing non-traditional security challenges, and deepening their cooperation in areas such as counterterrorism, cybersecurity, and infrastructure development. In addition, India and Japan should explore opportunities to engage with other regional actors, such as ASEAN nations, South Korea, and Australia, in trilateral and multilateral security dialogues and initiatives to collectively address regional security challenges. By considering these suggestions and recommendations, India and Japan can further strengthen their defense cooperation, contributing to regional

stability and security, and enhancing their ability to address common security challenges in the Indo-Pacific region.

## 5. Conclusions

In conclusion, the study found that India was initially reluctant to actively engage in anti-China security cooperation with Japan, but shifted its position towards a more proactive security partnership with Japan in the mid-2010s. The review of policy documents, political speeches, and government records reveals that India's reluctance until the mid-2010s can be attributed to its adherence to the principles of non-alignment and strategic autonomy, its desire to maintain a balance in its relations with both China and Japan, and domestic political considerations that prioritized economic development over military expansion or strategic alliances. Nevertheless, due to growing concerns over China's assertiveness and military modernization, a change in political leadership, as well as the development of the Quad as a platform for regional security cooperation, India has changed its attitude. It is a difficult challenge for the Indo-Pacific countries to curb Beijing's expansionism while maintaining good economic relations with China. Despite the prevailing perception that India-Japan defense cooperation is primarily driven by concerns over China's rising influence, the study highlighted that India has consistently pursued a policy of strategic autonomy and maintained a multi-vector approach to its foreign relations. Thus, the evolving India-Japan defense partnership should be seen as a natural progression of India's strategic partnerships rather than as an anti-China alliance.

To curve the growing assertiveness of China in the Indo-Pacific region, India and Japan should, on the one hand, deepen their cooperation in several ways, such as through defense technology collaboration, expanded military interoperability, and enhanced maritime security cooperation. At the same time, New Delhi and Tokyo need to continue to pursue balanced relations with other regional powers, including China, to avoid exacerbating regional tensions and encourage constructive dialogue on regional security issues. The deepening of this partnership has contributed to the maintenance of regional peace and the promotion of a rules-based order in the Indo-Pacific region. The recommendations outlined in this research not only provide a roadmap for India, Japan, and other Indo-Pacific countries that are bound by deep economic ties with China, but also help confront China's military threats to contribute to regional stability and security, enhance their ability to address common security challenges, and foster a more peaceful and prosperous region.

**Author Contributions:** Conceptualization, K. and T.S.; methodology, T.S.; formal analysis, K. and T.S.; resources, K.; writing—original draft preparation, K.; writing—review and editing, K. and T.S; supervision, T.S. All authors have read and agreed to the published version of the manuscript.

**Funding:** This research received no external funding.

**Institutional Review Board Statement:** Not applicable.

**Informed Consent Statement:** Not applicable.

**Data Availability Statement:** Not applicable.

**Conflicts of Interest:** The authors declare no conflict of interest.

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
