# Peer review of "The Rise of China and Evolving Defense Cooperation between India and Japan"

_socsci, doi:10.3390/socsci12060333_

Round 1

Reviewer 1 Report

Overall, this article provided a short survey of Japan-India defense cooperation in the context of the rise of China. The observations and findings are cogently presented, albiet, the paper does not enage much wiht IR theory scholarship that could be relevant; however, as a short piece this is not necessarily a flaw that requires addressing. 

The English is fine.

Author Response

1. Thank you. As the reviewer rightly observed, as the paper is a short piece, researchers decided not to bring IR theory scholarship.

Reviewer 2 Report

First, it's not enough that there was resistance or reluctance by India. It is because a formal alliance has adverse consequences. In addition, I think the article needs to also reflect upon Japan's national security and defense strategies in this section, even though now they are outdated. Japan in 2013 didn't really do security cooperation. It took many years for the environment to change to warrant the political risks associated with partnerships. I'm unsure of why this paper neglects Japan's perspective. The focus is too India-centric and dilutes your analysis.

Second, this paper clearly misses the forest for the trees on Indian security policy.  Under the heading, "India's Shift in Position and an Assertive Foreign Policy," there's no mention of the actual geopolitical conditions that led to the changes--LAC, Sri Lanka, maritime security, piracy, counter-terrorism and more. An examination of policy documents just gives you fragments of the full picture. 

With these deficits, and a lack of both an engagement with the academic literature on the subject as well as empirical data, this paper unfortunately needs substantial revision.  

The English level is acceptable. Check for formatting errors. 

Author Response

  1. As per the suggestion, we are adding some relevant arguments/evidence regarding the adverse consequences of a formal alliance.
  2.  As per the suggestions, some subsections are added to Japan's national security and defense strategies.
  3.  As per the suggestions, the section on "India's Shift in Position and an Assertive Foreign Policy," tried to engage with actual geopolitical conditions. 

Thank you for the valuable suggestions.

Reviewer 3 Report

I have reviewed this article about the increasing partnership between India and Japan, and I find it well-researched.

Although slightly inclined to the Indian side, the topic is adequate and necessary since there is little information on that matter and, in general, on the Indo-Pacific cooperation and activities. Besides, the article is well-crafted and rather scientific.

The authors could press their point further by investigating Japanese opinion and actions. Also, adding more graphs on the increasing expenditure allocated to such matters could prove beneficial for the article. I am curious about the future evolution of defense between two nations not prone to militarism in recent decades if we rule out the alleged support of India to the Tamil Tigers movement of Sri Lanka and the LTTE. 

However, the article, in summary, seems complete and thoroughly researched. Some dashes here and there would enhance the applicability of this article to near-future policies. 

The conclusions are consistent and offer the potential for expansion.

Author Response

  1. As per the review suggestions, Japanese opinions and actions are given more importance.
  2.  As per the review suggestions, the authors tried to incorporate graphs on related data.

Thank you.

Round 2

Reviewer 2 Report

Thank you for addressing my earlier concerns. Your revisions are impressive and it appears from the quality and substance of the changes that significant thought and preparation went into them. Consequently, I have no more concerns about this manuscript.